# Learning in Practice: Collaboration Is the Way to Improve Health System Outcomes

**DOI:** 10.3390/healthcare7030090

**Published:** 2019-07-09

**Authors:** Pieter J. Van Dam, Phoebe Griffin, Nicole S. Reeves, Sarah J. Prior, Bronwyn Paton, Raj Verma, Amelia Giles, Lea Kirkwood, Gregory M. Peterson

**Affiliations:** 1School of Medicine, College of Health and Medicine, University of Tasmania, Hobart 7000, Tasmania, Australia; 2School of Medicine, College of Health and Medicine, University of Tasmania, Launceston 7250, Tasmania, Australia; 3School of Medicine, College of Health and Medicine, University of Tasmania, Burnie 7320, Tasmania, Australia; 4Agency for Clinical Innovation, Chatswood 2067, New South Wales, Australia; 5School of Pharmacy, College of Health and Medicine, University of Tasmania, Hobart 7000, Tasmania, Australia

**Keywords:** project-based learning, industry partnership, healthcare, redesign, health service improvement, education, learning evaluation, blended learning

## Abstract

Evidence suggests that it is challenging for universities to develop workplace-relevant content and curricula by themselves, and this can lead to suboptimal educational outcomes. This paper examines the development, implementation, and evaluation of Australia’s first tertiary graduate course in healthcare redesign, a partnership initiative between industry and university. The course not only provides students with an understanding of person-centered sustainable healthcare but also the skills and confidence to design, implement, and evaluate interventions to improve health service delivery. Increasing students’ application of new knowledge has been through work-integrated learning, a pedagogy that essentially integrates theory with the practice of workplace application within a purposely designed curriculum. The specific aim of this study was to examine the outcomes of the course after two years, utilizing an anonymous online survey of graduates. Sixty-two graduates (48%) completed the survey. Kirkpatrick’s four-level evaluation model was used to analyze the data. The analysis revealed high satisfaction levels in relation to the course content and delivery. Through successful completion of the innovative course, students had increased their knowledge of health system redesign methods and, importantly, the ability to translate that knowledge into everyday practice. Graduates of the clinical redesign course reported that they had been able to transfer their skills and knowledge to others in the workplace and lead further improvement projects.

## 1. Introduction

As the demand for healthcare grows, pressure is increasing on health services to deliver quality, timely, and equitable care. Services must also have the capacity to innovate and continuously improve [1,2,3]. The need for organizations to innovate is driving a shift in the expectations of healthcare professionals, where quality improvement (QI) is viewed as a core competency. Health service redesign, or clinical redesign, focuses on QI in a more balanced and blended approach [4]. QI is now a part of many preregistration training programs in medical, nursing, and allied health disciplines [5,6,7,8]. Organizations have also developed in-house training programs to build capacity for health service redesign [9,10,11].

QI education programs vary greatly from preregistration, single-discipline, university-based programs that focus on theory and knowledge acquisition [12] to large interprofessional project-based programs that aim to directly improve care and patient outcomes. Despite the range of programs offered, there is some agreement in the literature about key characteristics of effective QI education [8,13,14]. Successful components include an interprofessional learner cohort, a pedagogy that combines didactic theoretical learning with experiential, project-based learning and coaching. In their review of QI education, however, Starr et al. [8] noted a decrease over time in the number of studies that described programs which included coaching and interprofessional learning. This may reflect the complexity and additional resourcing required to integrate these elements [8].

One solution to address the complexities and resource requirements of delivering a comprehensive QI curriculum is for healthcare organizations to partner with universities. For universities, partnering with organizations provides students with the opportunity to develop work-ready skills by undertaking workplace projects, in which theoretical knowledge is put into practice. Organizations benefit from access to relevant academic expertise and sharing resources required to deliver the program. Students benefit from a structured curriculum in which the achievement of learning outcomes is measured at an individual level through formal assessment and feedback [15,16,17]. Students also have their learning recognized by being awarded a tertiary qualification.

In this paper, we describe the development of a partnership between the Agency for Clinical Innovation (ACI), a pillar organization of New South Wales Health (NSW Health) in Australia and the University of Tasmania (UTAS) to deliver a postgraduate health service redesign course. The Graduate Certificate (Clinical Redesign) aims to build capacity both at the individual student and system level to achieve the Institute of Healthcare Improvement’s Triple Aim through health service redesign [18]. The course commenced in 2016 and is open to all health professionals employed by NSW Health. It combines didactic delivery with workplace project-based learning and on-the-ground coaching. We also report the findings from an evaluation of the course’s effectiveness using Kirkpatrick’s four-level model of training evaluation [19,20].

The NSW ACI partnered with UTAS to deliver their 12-month Redesign Program as a postgraduate qualification. ACI had been offering the course for almost 10 years as a vocational education and training diploma. ACI and UTAS content was combined to produce a Graduate Certificate award course that is practically applied in the workplace, delivered in a blended learning model, and meets Australian Qualification Framework (AQF) Level 8 requirements. The pre-existing ACI course was particularly strengthened by providing knowledge of research principles and methods. In the Graduate Certificate (Clinical Redesign), students learn by doing; the course requires students to identify and undertake a workplace improvement project using Healthcare Redesign Methodology [21]. Projects are designed to meet the IHI’s Triple Aim [18] and are chosen to align with students’ individual health service strategic priorities. This ensures that energy and resources are spent addressing areas of need and that projects deliver sustainable changes that benefit patients and the wider population and reduce the cost of care.

The Graduate Certificate (Clinical Redesign) is coordinated and delivered in a partnership between ACI and UTAS. The course structure and overall responsibilities for delivery were formalized in a Memorandum of Understanding. The agreement provides students with structured learning pathways situated in a Work Integrated Learning (WIL) context. The pathway permits students to undertake three UTAS units in Sydney, delivered as equivalent units by Adjunct staff of the UTAS. A fourth unit is delivered fully online. The Graduate Certificate (Clinical Redesign) is targeted at both clinical and nonclinical healthcare staff, including all levels of clinicians from medical, nursing, scientific and allied health teams, senior and middle healthcare managers, and business administration staff working in the healthcare industry. The course is the first of its kind in Australia and bridges the gap between academia and practice by enabling students to learn by delivering real-world benefits through work-integrated learning while achieving a postgraduate qualification. The course incorporates a mixed-method model of delivery to enable NSW Health staff to complete real-time, workplace clinical redesign projects. The model of delivery includes theoretical components in which students are required to fulfill the academic areas of the course, as well as a practical component designed to address the needs of their organizations. The curriculum is centered on practice, using experiential learning, problem solving, and the analysis of practice within theoretical frameworks and the assessment of evidence [22].

ACI course staff worked with UTAS academic staff to develop course and unit-level intended learning outcomes, administrative systems, timelines, marking processes, and assessment rubrics. Two fundamental guiding principles were developed: (i) to retain the work-integrated blended learning model; (ii) to establish outcomes that enabled attainment of the required capabilities to lead improvement initiatives [23]. This ensures that the course meets academic requirements while producing highly competent health service improvement practitioners who deliver projects contributing to better patient and organizational outcomes, in line with the principles of Triple Aim and clinical redesign.

### Course Structure

There are three course intakes per year with an interactive curriculum. The overall learning objectives can be found in Table 1.

The course consists of five stages of redesign methodology (Figure 1) integrating a range of tools to develop robust improvement projects. This methodology focuses on developing solutions that address the right root causes and provides students with the required skills to implement change and achieve sustained outcomes. Using evidence-based redesign methodology [24,25], the course teaches healthcare professionals how to identify the root causes of issues impacting patient journeys, and then develop and implement sustainable change processes to improve the way healthcare is delivered. Interprofessional student teams may include nursing, allied health, medical, and administrative personnel.

All projects are supported with workplace coaching from ACI redesign leads and project sponsors in Local Health Districts/Specialty Health Networks, as well as the academic teaching team at UTAS. The first components of the course (initiation, diagnostics, and solution design) take five months in total and require a considerable time commitment. Students are expected to be backfilled in their employment for 2–3 days a week, dependent on the scope of their project. During this time, senior health staff help students with their health service improvement projects. These staff members (redesign leads) have been employed as adjunct lecturers (*n* = 15) with UTAS. This practice ensures that industry partners are closely involved in aspects of the course, such as feedback, design, content development, and assessment. Redesign leads are experts in redesign, and project and change management, and are responsible for identifying potential projects and students within their organization, and also providing mentoring and coaching for local teams throughout the conduct of the project. The project sponsor is an individual at a senior or executive level who has completed the course and can authorize change and play a key role in delivering and sustaining the implementation.

A final online unit, Translational Research and Health Service Innovation (CAM538) is delivered entirely by UTAS via an online learning platform, MyLO, and completes the Graduate Certificate qualification. Students who choose not to complete this unit are eligible to graduate with a Certificate of Achievement from ACI only. The translational research unit provides a framework for understanding the links between knowledge and practice, and the barriers and enablers for successful implementation of this knowledge. This unit aims to encourage the development of the capacity of health service leaders to identify and evaluate emerging knowledge relevant to their practice and to implement change based on this knowledge within their sphere of influence.

Students address numerous constraints and issues inherent in health service delivery across acute, subacute, and community health services, such as protracted wait times, care coordination and integration, access and referral across services, and timely treatment and discharge. Reduction of waste, duplication, and inefficiencies underpin the approach. This leads to the realization of benefits such as released capacity, cost reduction and avoidance, improved quality and patient safety, and, ultimately, the more effective use of finite resources.

## 2. Materials and Methods

A largely quantitative research approach has been used to evaluate the course, aiming to assess course outcomes for impact and effectiveness two years after the introduction of the new course. The study used a questionnaire combining Likert Scale, multiple-choice, and open-ended questions. The questions were adapted from previous validated surveys aimed at undergraduate and postgraduate students completing university study [26,27]. The 23-item, online survey (SurveyMonkey) was used to ask graduates about their experience with the Graduate Certificate (Clinical Redesign) course, outcomes since graduation, and the partnership between ACI and UTAS. The questions were largely based on the intended learning outcomes. Kirkpatrick’s Model, Four Levels of Learning Evaluation [1], was employed to evaluate the results. This four-level model consists of reaction, learning, behavior, and results criteria, and provides feedback about the effectiveness of the effort to serve numerous stakeholders.

Graduates who had completed the course since its inception in 2016 were identified through the ACI database and emailed a letter of invitation and web link to the anonymous survey. Each participant was requested to read the information sheet at the beginning of the survey and consent was assumed by submission of the survey. The survey was open for eight weeks between 1 September 2018 and 31 October 2018. Ethical approval was obtained from the Tasmanian Health and Human Research Ethics Committee (H0017402). SPSS version 24 (IBM Inc., Chicago, IL, USA) was used to analyze the data.

## 3. Results

One hundred and forty-five students completed the clinical redesign course in 2016 and 2017, with 130 (90%) students completing the Graduate Certificate. The remaining 15 (10%) students completed the ACI certificate of achievement. Sixty-two graduates of the Graduate Certificate course (48%) completed and submitted the online survey, with the sample demographics shown in Table 2.

### 3.1. Graduate Levels of Satisfaction (Q5, Q7, Q11, Q14)

The first level in Kirkpatrick’s Model addresses the criteria *reaction*, which involves the thoughts and feelings of the participants in relation to the course. Overall, the levels of satisfaction were high, with 49 (80%) participants strongly agreeing that they were satisfied overall with the course. Participants strongly agreed (82%) that the course made a worthwhile contribution to their professional development. The full results are shown in Table 3. The free-text comments relating to the benefits gained from the course suggested that participants were satisfied that the course provided beneficial content and delivered on the overall learning objectives:(a)“A clear applicable understanding of redesign methodology and translational research;”(b)“Structured methodology for implementing improvement and healthcare change projects;”(c)“Confidence in pursuing change in the workplace;”(d)“An improved awareness of the shift required to ensure evidence becomes practice.”

The majority of participants strongly agreed that the course contained a good mix of theoretical and practical knowledge (74%), was helpful for building stronger networks for health service improvement (65%), supported them to undertake their projects (74%), and encouraged patient-centered thinking (81%). The comments relating to course structure showed that the course was well received overall; however, some components differed between students.
(a)“I liked the combination of theoretical and practical. It is a very well supported course. The timelines were tight, but I found that kept me on track;”(b)“Face to face days were excellent and well presented;”(c)“The timeframes for completing can be challenging due to the nature of decision making in health, access to key staff required to inform project (busy clinicians, lots of staff on leave constantly).”

### 3.2. Graduate Learning (Q16)

Learning is the focus of the second level of Kirkpatrick’s Model, with a focus on the improvement in knowledge and skills, and changes in attitudes. Participants strongly agreed that the course had increased their capacity to consider others’ frames of reference to generate engagement and buy-in (84%), to look at different perspectives and reflect on their own practice (70%), and to identify issues in the workplace and build a case for change (82%).

### 3.3. Graduate Use of Knowledge and Skills (Q17)

In level three of Kirkpatrick’s Model, behavior changes are evaluated through how well students have been able to apply their learning in their workplace. Question 17 addressed the use of skills and knowledge from the course and its integration into the graduate’s workplace. Thirty-seven (67%) of the graduates surveyed strongly agreed that they have often used the skills and knowledge that they acquired in the course. Twenty-three (43%) participants felt their perspective was more patient-centered after completing the course, and 27 (50%) participants have led further health service improvement projects.

### 3.4. Outcomes of Graduate Learning (Q19)

The fourth level of Kirkpatrick’s Model focuses on results and how outcomes are achieved for the organization, and how projects have been shared. Participants had shared their project at a conference (44%), had others view their project (40%), or had their solutions adopted elsewhere in their organization (39%). The remaining participants had their project outcomes adopted by other organizations (24%) or published (8%). Only 8% of participants had not had their project shared. Twelve participants selected ‘other’ as their response and the free-text field indicated that some of the projects had actually been shared or implemented with other Local Health Districts or state-wide.

## 4. Discussion

The course design, structure, and delivery were well received by graduates and, more importantly, partnering industry with academia has proven to be successful in enabling students to meet course learning outcomes, leading to transfer of learning [28]. Students were satisfied with the course and would recommend the course to others. Students improved their knowledge and increased skills in healthcare redesign as a result of completing the course [20,29]. In carrying out their projects, students questioned their own problem-solving approaches, making them active learners, and this is required to produce change [30]. Learning involves a process of a relatively permanent change in behavior as a result of experience [31]. Students had changed their behaviors, as illustrated by undertaking further projects [1,29]. Finally, students had achieved results beyond the intended learning outcomes through the adoption of their project elsewhere within and outside their organization [20,29]. It appears that the four levels of evaluation, as specified by Kirkpatrick’s Model of training evaluation [20], have been met.

It is clear that students have developed their cognitive skills, but also relevant abilities impacting on healthcare organizations, while enhancing their learning outcomes [32]. The project-based approach, with a focus on contextual variety and situated practice, provides a suite of opportunities for experiential learning events and contextual experiences in which learners engage [33]. Therefore, simultaneously meeting the course learning outcomes and successfully completing a broad range of initiatives might be directly related to a well-supported, project-based learning design [34]. Support and constructive feedback in project-based learning is of vital importance for building confidence, learning, retention, and commitment [35].

Another factor at play was the strengths that each party brought to the partnership and the genuine willingness to work together. However, to make a partnership work, each side must overcome the challenges it creates [36]. At the start of the course, challenges were related to different ways of working between UTAS and ACI in delivering education and training [34]. However, these early challenges did not seem to have impacted the students’ learning journey. While both UTAS and ACI engage in education, they have distinct objectives and are founded on different business models [37]. The objectives from both organizations were met by having an underlying shared teaching philosophy, that the best way to learn health service improvement is by doing [6]. The alignment of course and unit intended learning outcomes assisted with creating a seamless codelivered course. Constructively aligned courses encourage students to engage in a deep learning approach and discourage students from surface learning activities [23], and is particularly useful for project-based learning [38].

Codevelopment of assessment criteria, marking rubrics, and the creation of a rigorous moderation process to ensure feedback and marks were fair and appropriate assisted in developing a rigorous health service improvement course. Authentic assessment in higher education is defined as activities “that engage students in real-world inquiry problems involving higher order thinking skills with an authentic audience beyond the classroom” [23]. This permits a holistic rather than a fragmented approach and this might explain the high scores in the outcomes achieved, based on Kirkpatrick and Kirkpatrick’s [20] level four. Quarterly steering committee meetings, weekly operational teleconferences, and travel for key events and meetings helped with strengthening the partnership through the development of strong and authentic relationships. A joint position to support course administration ensured that the course ran smoothly and protected teaching time.

Approaches to process improvement, such as Plan-Do-Study-Act (PDSA) and Project Management practice, are often sufficient for operational improvement initiatives in care delivery processes [39]. From a higher education and Australian Qualification Framework (AQF) perspective, a focus on a solid research approach is required to utilize the best evidence available [40]. The inclusion of a translation research/implementation science unit has provided students with the transformational leadership skills and knowledge required to address evidence-practice gaps, as well as process redesign problems. The results suggest that graduates used their knowledge gained from the translational research unit in assisting with developing strategies to overcome implementation barriers. The focus on implementation science in the Graduate Certificate complements the redesign skills that students have gained through their workplace projects. This allows students to become more rounded healthcare improvement experts, capable of addressing process redesign problems, as well as evidence-practice gaps.

Previous training courses in the arena of health service improvement have often been shown to be disconnected from the clinical space and difficult to apply [41]. The clinical redesign course has shown that placing emphasis on supporting students within their workplaces has positive outcomes in students successfully completing the course. There is relatively little written about the benefit of project-based learning and partnering with industry [34]. However, it is clear that partnerships can provide opportunities to generate new knowledge and translate that knowledge into healthcare redesign initiatives [36]. The course’s unique collaboration between government, workplace, and academia presents opportunities for continued learning across each sector, as well as future transfer of skills and knowledge within the healthcare space.

Several limitations have been identified pertaining to this study. The first relates to the timeline for Kirkpatrick and Kirkpatrick’s [20] level 4 results. The study reports on the perspectives of graduates who started the course in 2016 and 2017. A complete understanding of the impact on project implementation often takes a few years, particularly in relation to sustainability. The year 2016 could be seen as transition period, whereby both UTAS and ACI had to design solutions for different ways of working and therefore this may not reflect current student experience. The survey response rate of almost 50% was acceptable, although higher would have been clearly preferable and more likely to provide generalizable results. The survey was self-reported and the level 4 results may not reflect the organizations’ and executive sponsors’ perspectives. Further research will need to be undertaken to collect data from these cohorts.

Boonyasai et al. [14] published the first systematic review of health service improvement curricula for healthcare professionals. Their review highlighted that students’ knowledge, attitudes, and involvement in health service improvement increased by employing a variety of teaching methods, but few studies have measured the outcomes on patients and healthcare organizations. The authors suggested that measuring patient and organizational outcomes would add a rich dimension to the understanding of the impact of partnership courses beyond the classroom.

## 5. Conclusions

Concluding, the course has evolved beyond teaching theory and knowledge, and has adopted new and innovative approaches, working closer with the healthcare industry. Combining tertiary education with contextual work-integrated learning, designed for and with healthcare organizations, is an important investment. This approach is essential to ensure that the skills health students develop are well matched to the complex and changing healthcare industry.

## Figures and Tables

**Figure 1 healthcare-07-00090-f001:**
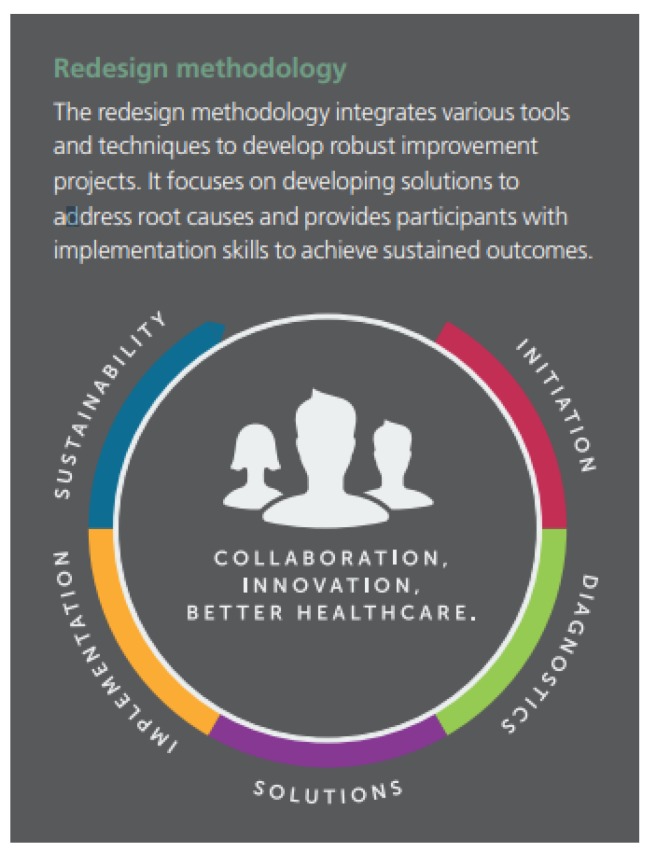
Redesign Methodology.

**Table 1 healthcare-07-00090-t001:** Overall Learning Objectives for the Course.

Domain	Specific Learning Objective
Knowledge	Specialized, systematic, and coherent knowledge that applies a system-wide and multidisciplinary approach to resolving Health Services and Healthcare improvement issues.
Evidence-Based Practitioner	Cognitive skills to independently think critically and review, analyze, consolidate, and synthesize knowledge, and generate and evaluate solutions to complex health services and health improvement issues.
	Comprehensive knowledge of the range of research principles, methods, and frameworks applicable to health systems improvement within organizations.
Communication	Skills that facilitate transfer of complex knowledge and ideas to a variety of audiences.

**Table 2 healthcare-07-00090-t002:** Demographic data for survey sample.

Variable	Number of Responses *
**Gender**	
Male	13 (21%)
Female	49 (79%)
**Highest Level of Education**	
Diploma/Certificate	5 (8%)
Undergraduate degree	10 (16%)
Postgraduate degree	45 (73%)
Other	2 (3%)
**Professional Background**	
Nursing	34 (55%)
Allied Health	14 (23%)
Medical	3 (5%)
Administration/Support	3 (5%)
Other	6 (10%)

* Not all participants answered every question; * *n* = 62.

**Table 3 healthcare-07-00090-t003:** Results of online survey.

Question Number	Results				
	Strongly Agree	Slightly Agree	Neutral	Slightly Disagree	Strongly Disagree
**Graduate levels of satisfaction**					
Kirkpatrick Level 1 (Reaction, thought, and feelings)					
5. *Overall levels of satisfaction with the Redesign Program*:					
5a. Overall, I was very satisfied with the Program	49 (80%)	12 (20%)	-	-	-
5b. The Program has made a worthwhile contribution to my professional development	51 (82%)	11 (18%)	-	-	-
5c. I would recommend the Program to others	49 (79%)	12 (19%)	1 (2%)	-	-
7b. The Program contained a good mix of theoretical and practical knowledge	46 (74%)	14 (23%)	2 (3%)	-	-
7c. I found the Program helpful for building stronger networks for health service improvement initiatives	40 (65%)	15 (24%)	7 (11%)	1 (2%)	-
7d. The Program design supported students to undertake their projects	46 (74%)	10 (16%)	3 (5%)	3 (5%)	-
7e. The Program encouraged patient-centered thinking	50 (81%)	9 (15%)	3 (5%)	-	-
11e. Facilitators worked with my organization to support my learning	41 (67%)	12 (20%)	6 (10%)	1 (2%)	1 (2%)
11f. The program assessments contributed to my learning	49(79%)	13 (21%)	-	-	-
14e. Translational Research Unit complemented the ACI component of the Program	44 (71%)	11 (18%)	4 (6%)	3 (5%)	-
**Graduate learning**					
Kirkpatrick Level 2 (Learning—changes in knowledge, skills, attitudes)					
18. *The Program has increased my capacity to:*					
a. Look at things from different perspectives and reflect on my own practice	49 (79%)	13 (21%)	-	-	-
b. Identify workplace issues and use data/evidence to build a case for change	51 (82%)	10 (16%)	1 (2%)	-	-
c. Consider others’ frames of reference to generate engagement and buy-in	52 (84%)	10 (16%)	-	-	-
d. Work with others to facilitate change in my organization	51 (82%)	10 (16%)	1 (2%)	-	-
e. Evaluate changes made to health service delivery	42 (68%)	18 (29%)	1 (2%)	1 (2%)	-
f. Make changes to deliver more patient-centered outcomes	47 (76%)	12(19%)	2 (3%)	1 (2%)	-
**Graduate use of knowledge and skills**					
Kirkpatrick Level 3 (Behavior—how on-the-job training has changed as a result of the learning)					
19. *Use of skills and knowledge*					
a. I have often used the skills and knowledge acquired in the Program	37 (67%)	16 (29%)	1 (2%)	1 (2%)	-
b. I have led further health service improvement projects	27 (50%)	15 (28%)	4 (7%)	5 (9%)	3 (6%)
c. I have been able to mentor others in my workplace	25 (50%)	20(40%)	5 (10%)	-	-
d. Completing the Program has helped me gain a promotion	10 (19%)	15 (28%)	14 (26%)	6 (11%)	8 (15%)
e. I feel my perspective is more patient-centered	23 (43%)	17 (31%)	13 (24%)	1 (2%)	-
**Outcomes of graduate learning**					
Kirkpatrick Level 4 (Results—the outcomes achieved for the organization as a result of learning)					
22. *How has your project been shared? (May select more than one answer):*					
My solutions have been adopted elsewhere in my organization	24 (39%)				
My solutions have been adopted by other organizations	15 (24%)				
a. I have presented my project at a conference	27 (44%)				
b. The outcomes of my project have been published	5 (8%)				
c. Others have come to see my project	25 (40%)				
d. Other: (Free text field)	12 (19%)				
e. Unsure	9 (15%)				

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
