# Peer review of "Learning in Practice: Collaboration Is the Way to Improve Health System Outcomes"

_healthcare, 2019, doi:10.3390/healthcare7030090_

Round 1
Reviewer 1 Report
i have reviewed the artticle and found it well written, articulated, and the research concept and methodology are well allined and defined, the research do support the fact that investment in Health System through improving the knowledge of the Health workers play an essential role in uograding the health services through tqualified and well trained health workers.
The article strongly recommended for publication
Author Response
We would like to thank the reviewer for their thoughtful comments about our manuscript Learning in practice: collaboration is the way to improve health system outcomes.
The manuscript has been thoroughly edited.
Reviewer 2 Report
1The paper uses anonymous online survey to examine the outcomes of the course after two years experiment, which is sound and logic.
2 The author uses Kirkpatrick’s four-level evaluation model to analyze data is experimental and correlative to quantitative process.
3 The conclusion is innovative and enlightening.
1-Sixty-two graduates (48%) completed the survey is not enough to be the experimental sample. 2-Kirkpatrick’s Model will be introduced in theoretical background. 3-The language in abstract and conclusion can be polished.
Author Response
We would like to thank the reviewer for their thoughtful comments and suggestions for improving our manuscript Learning in practice: collaboration is the way to improve health system outcomes.
In response to concerns about the small sample, we believe that a response rate of approximately 50% can be considered reasonable, especially given the two-year follow-up period. Nevertheless, we have added the following within the limitations section at the end of the manuscript: The survey response rate of almost 50% was acceptable, although higher would have been clearly preferable and more likely to provide generalisable results.
We have added the following to the methods section to describe Kirkpatrick’s Model in more detail: Kirkpatrick’s Model, Four Levels of Learning Evaluation[1], was employed to analyse and evaluate the results of the graduate outcomes survey. This four-level model consists of reaction, learning, behavior and results criteria, and provides multilevel feedback about the effectiveness of the effort to serve numerous stakeholders.
The manuscript has been thoroughly edited and the language in the abstract and conclusion refined.
Reviewer 3 Report
The study evaluated educational outcomes associated with new course delivery using Kirkpatrick’s model. The manuscript is well-written, I have only minor suggestions.
The study used previously validated survey - I wonder if this survey was developed using Kirkpatrick’s model and validated, or the authors simply framed the existing survey (not developed under Kirkpatrick’s framework) using Kirkpatrick’s model for this research. If the latter is the case, I would recommend reporting reliability and validity statistics in order to interpret the results in a way that "the four levels of the model have been met". If the authors categorized the questions in four concepts (levels) on their own, they need to present an evidence that their categorization is correct, meaning questions represent the concepts well. If the former is the case, I would recommend reporting reliability and validity statistics from the study that developed the questionnaire. If these statistics are not available, the study should not conclude that all four levels have been met.
Since Kirkpatrick’s model is a major framework for the analysis, I would have a separate section explaining the model under introduction section for the readers who are not familiar with the model. In this section, the authors could report the reliability and validity evidence.
Most responses were highly skewed to strongly agree/agree with little variation, suggesting ceiling effect. How did you address this concern?
Author Response
We would like to thank the reviewer for their thoughtful comments and suggestions for improving our manuscript Learning in practice: collaboration is the way to improve health system outcomes.
The questionnaire used in the study was not validated. The questions were adapted from previously reported validated questionnaires and designed to align with the course intended learning outcomes. Kirkpatrick’s model was employed to evaluate the results. The description of how the questionnaire was developed in the methods has been refined. We have also softened the language in the discussion regarding whether all four levels of Kirkpatrick’s Model have been met: It appears that the four levels of evaluation, as specified by Kirkpatrick’s Model of training evaluation[1], have been met.
We have added the following to the methods section to describe Kirkpatrick’s Model in more detail: Kirkpatrick’s Model, Four Levels of Learning Evaluation[1], was employed to analyse and evaluate the results of the graduate outcomes survey. This four-level model consists of reaction, learning, behavior and results criteria, and provides multilevel feedback about the effectiveness of the effort to serve numerous stakeholders.
We may have misunderstood the reviewer’s comments about a ceiling effect, but we don’t believe there is a real concern. The options included “strongly agree” and “slightly agree”, which are quite distinct options (as opposed to “strongly agree” and “agree”). The relatively high rates of “strongly agree” should be seen as a positive finding i.e. a strong and consistent endorsement of graduates’ experiences of the course.
The manuscript has been thoroughly edited.